# Advances in Therapeutic _L_-Nucleosides and _L_-Nucleic Acids with Unusual Handedness

**DOI:** 10.3390/genes13010046

**Published:** 2021-12-24

**Authors:** Yuliya Dantsu, Ying Zhang, Wen Zhang

**Affiliations:** 1Department of Biochemistry and Molecular Biology, Indiana University School of Medicine, 635 Barnhill Drive, Indianapolis, IN 46202, USA; yudantsu@iu.edu (Y.D.); yzh11@iu.edu (Y.Z.); 2Melvin and Bren Simon Cancer Center, 535 Barnhill Drive, Indianapolis, IN 46202, USA

**Keywords:** nucleic acid therapy, _L_-nucleoside analogue, _L_-aptamer, nanoparticle

## Abstract

Nucleic-acid-based small molecule and oligonucleotide therapies are attractive topics due to their potential for effective target of disease-related modules and specific control of disease gene expression. As the non-naturally occurring biomolecules, modified DNA/RNA nucleoside and oligonucleotide analogues composed of _L_-(deoxy)riboses, have been designed and applied as innovative therapeutics with superior plasma stability, weakened cytotoxicity, and inexistent immunogenicity. Although all the chiral centers in the backbone are mirror converted from the natural _D_-nucleic acids, _L_-nucleic acids are equipped with the same nucleobases (A, G, C and U or T), which are critical to maintain the programmability and form adaptable tertiary structures for target binding. The types of _L_-nucleic acid drugs are increasingly varied, from chemically modified nucleoside analogues that interact with pathogenic polymerases to nanoparticles containing hundreds of repeating _L_-nucleotides that circulate durably in vivo. This article mainly reviews three different aspects of _L_-nucleic acid therapies, including pharmacological _L_-nucleosides, Spiegelmers as specific target-binding aptamers, and _L_-nanostructures as effective drug-delivery devices.

## 1. Introduction

The genetic information of the human genome is stored in nucleic acids, which are constituted by only four major nucleobases (A, G, C, and T). DNA replication and transcription, as well as RNA translation, are specifically regulated by proteins recognizing nucleic acids [1]. In order to preserve the integrated genomic information and secure the accurate nucleic acid-protein recognitions, native DNA and RNA molecules must bear not only the particular base sequence but also the highly ordered structures, including phosphodiester linkages, (deoxy)ribose moieties, and a variety of unique secondary/tertiary geometries [2,3]. Like the phenomenon that most chiral components of natural products occur predominantly in one of the plausible enantiomeric isomers, the biological building blocks of life, including nucleic acids and proteins, are exclusively utilizing only one chirality to function in the metabolism [4]. In particular, native DNA and RNA are only composed of _D_-configuration, while native proteins only involve _L_-amino acid residues, except for the rarely arisen _D_-amino acids and _D_-peptides [5,6]. In theory, the two enantiomeric DNAs/RNAs should have identical chemical and physical properties; nevertheless, all living organisms on the earth only utilize _D_-DNAs/RNAs. Meanwhile, only _L_-proteins can contact _D_-DNAs/RNAs in vivo to ensure the central dogma [7]. This scenario would be closely related to the molecular evolution of life. There is no substantial solution to this evolutionary mystery yet, but it is believed to relate with some nucleoside- or nucleoside-like-molecule-involved prebiotic chemistry that occurred in the primordial soup on the early earth, catalyzed by metal ions, nucleic acids, amino acids, or other coenzymes [8].

In this article, we are not trying to seek the certainty that the homochirality of natural nucleic acids agrees with or diverges from the classic RNA world hypothesis in evolution [9]. We are interested in another actuality: _L_-DNAs/RNAs, as the completely orthogonal biomolecules of native nucleic acids, are featured due to their resistance to nuclease degradation, nontoxicity, and nonimmunogenicity [10]. Those are the properties an ideal nucleic acid drug is expected to possess. Therefore, there has been continuous active research in this field. Importantly, _L_-DNAs/RNAs are incapable of base pairing with native nucleic acids in vivo or interacting with native nucleic-acid-processing enzymes. This precludes the possibility of using _L_-nucleic acids such as antisense therapy [11] or a siRNA strand to induce Dicer-involved RNA interference [12]. The current application of _L_-nucleic acids as therapeutic agents in clinical trials mainly focuses on the _L_-aptamer development, and it utilizes the versatile secondary/tertiary structures of _L_-nucleic acids to recognize the disease relevant target with high specificity and affinity. The functional _L_-aptamer is discovered from the combinatorial library through in vitro SELEX (systematic evolution of ligands by exponential enrichment) [13,14]. Other successful applications of _L_-nucleic acid drugs include the construction of an entire _L_-DNA nanoparticle to deliver therapeutic ligands, and the usage of an _L_-nucleoside with a history of 60 years [15]. This review will mainly discuss three aspects of _L_-nucleic acids, including _L_-nucleoside, _L_-aptamer, and _L_-nanoparticle, and introduce the instances closely related to disease treatment. Other biotechnological applications of _L_-nucleic acids, such as biosensors [16], are beyond our emphases in this article. Those related contents have been summarized in other review articles [17,18].

## 2. Nucleoside Analogs as Therapeutic Antiviral and Antitumor Agents

Nucleoside analogues have been playing a vital role as a critical chemotherapy of viral infectious diseases [19,20]. Due to the conformational similarity with naturally occurring nucleotides/nucleosides, various carbocyclic nucleoside analogues have been designed, which can specifically recognize the target polymerases [21] or hydrolases [22] and effectively block their biological activities. Particularly, the outbreaks of severe acute respiratory syndrome-coronavirus-2 (SARS-CoV-2) in 2019 have especially raised the concerns about the deficiency of effective therapeutics and highlighted the importance of developing multiple antiviral strategies due to the fast drug-resistant mutations. Therefore, tremendous efforts have been explored to develop the broad-spectrum nucleoside analogues targeting the SARS-CoV-2-RNA-dependent RNA polymerase [23]. However, the application of wild-type carbocyclic nucleoside analogues is frequently restricted by the arisen cytotoxicity, after the 5′-phosphorylation happens to the nucleoside analogue and this metabolite interferes with various normal cellular enzymes [24]. To seek a novel platform to diminish the unwanted toxicity, the enantiomeric nucleoside analogues containing the unnatural _L_-configuration have been pioneered. We will briefly discuss the current advances in _L_-nucleoside analogues as selective antiviral ligands as well as the corresponding structure-activity relationship studies.

### 2.1. _L_-Type Neplanocin Compounds as Anti-Norovirus Therapies

9-(trans-2′,trans-3′-dihydroxycyclopent-4′-enyl)-3-deazaadenine (DHCDA) is a neplanocin A analog and functions as an inhibitor of S-adenosylhomocysteine (AdoHcy) hydrolase, with broad-spectrum antiviral potential against vesicular stomatitis virus, vaccinia virus, parainfluenza virus, reovirus, and rotavirus [25]. Inspired by the observations that both _D_-like and _L_-like neplanocin derivatives possess potent antiviral activities, Chen lab has pioneered to synthesize a series of DHCDA analogues, all of which contain _L_-type configuration and an adenine nucleobase but lack the 5′-hydroxyl group for nucleoside kinases to recognize [26].

The synthesized _L_-neplanocin derivatives contained a cyclopentenol pseudo-sugar and an adenine nucleobase analogue, which were coupled via a Mitsunobu coupling reaction to provide the target compounds (Figure 1). The syntheses of halogenated derivatives, including bromonation at its nucleobase and 5′-fluorination/bromonation at sugar, were also accomplished. These _L_-like analogues of natural carbocyclic nucleoside neplanocin A were evaluated as potential inhibitors against norovirus and Ebola viruses. Compounds **1a** and **2b** showed potent antiviral activity against norovirus (EC_50_ = 4.2 µM and EC_50_ = 3.0 µM, respectively). Compound **1b** was found to possess effective antiviral activity against the Ebola virus (EC_50_ = 8.3 µg/mL), while the analogues **2a** and **2b** were completely inactive. With the additional exocyclic derivatizations at the 5′-position, compound 5 displayed the potent activity against Pichinde (EC_50_ = 0.9 µg/mL) and Tacaribe (EC_50_ = 1.3 µg/mL). Both viruses are negative single-stranded RNA viruses belonging to arenaviridae, the same family as the Lassa fever virus. Compound **4a** and compound **4b** were also active against the Ebola virus. Although **4b** showed no activity against those two arenaviruses, it is two-fold more active against the Ebola virus than compound 5.

It is noteworthy that the conformational behavior of “sugar” puckering (north/south) and nucleobase orientation (syn/anti) may contribute to the antiviral activity differences. The crystallographic studies revealed that the sugar in compound **1b** adopted a 3′-exo conformation, while the more conformational, rigid bicyclic sugar in compound **2b** would lock it into 2′-exo conformation. In addition, because of the steric hindrance, the 3-bromo substitution played an important role in anti-Ebola activity by forcing the nucleobase to adopt the less congested anti-conformation over the syn-conformation. No activity was found for single-stranded positive viruses, which suggests that _L_-like neplanocin analogues are more effective on single-stranded negative-sense RNA viruses [27].

### 2.2. _L_-Enantiomer of Immucillin Analogue as an Anti-T-Cell Leukaemia Agent

Immucillin-H (ImmH, also known as Forodesine) is a transition-state analog inhibitor of purine nucleoside phosphorylase (PNPases). It has been extensively studied for the treatment of patients with T-cell acute lymphoblastic leukemia (T-ALL), and its C-nucleoside hydrochloride form is in phase II clinical trials as an anti-T-cell leukemia agent [28]. Various Immucillin analogs modified at the 2′-, 3′-, or 5′-positions of the aza sugar moiety or at the 6-, 7-, or 8-positions of the deazapurine, have been synthesized and tested for their inhibition of human PNPases [29]. Inspired by the nucleoside-like structures of Immucillin analogues and their binding modes with PNPases by crystal structures [30,31], their _L_-enantiomers have been investigated as novel pharmaceuticals against T-cell mediated disorders [32]. The synthetically achieved (1R)-1-(9-Deazahypoxanthin-9-yl)-1,4-dideoxy-1,4-imino-_L_-ribitol (Figure 2, **6a**) was an _L_-enantiomer of natural _D_-ImmH, and its hydrochloride complex was revealed to be a slow-onset tight-binding inhibitor of PNPases of human, bovine, and *Plasmodium falciparum.* Although this compound showed less activity than _D_-ImmH when inhibiting the selected enzymes, it still demonstrated more excellent binding potency compared to 3′- and 5′-modified _D_-ImmH. Moreover, the _L_-enantiomer of second-generation Immucillin analogue, 4′-deaza-1′-aza-2′-deoxy-1′-(9-methylene)-Immucillin-H (DADMe-ImmH) [33], was also synthesized. The _L_-DADMe-ImmH (Figure 2, **6b**) also displayed lower activities as an inhibitor, when binding to the three enzymes. However, it was interesting to observe the sub-nanomolar binding capacities of these two _L_-formed Immucillin analogues, plus they had the potential to be applied in different circumstances.

### 2.3. _L_-d4T and _L_-ddC Derivatives as Anti-HIV Agents

There have been extensive studies in search of chemotherapeutics to effectively cure pathogenic Human Immunodeficiency Virus (HIV) [34]. The dominant experimental and clinical attempts are focused on the development of modified nucleoside analogues, which bind to HIV reverse transcriptase and interfere with the synthesis of DNA copying of the viral genome [35]. Successful examples include small molecule drugs of AZT [36], ddI [37], ddC [38], and d4T [39] that have been clinically used to treat AIDS patients. The lack of 2′- and 3′-hydroxyl groups in nucleoside sugar can cause the termination of HIV reverse transcription and inhibit the viral life cycle. In particular, d4T has the double bond in its pseudosugar ring to rigidify the ring to planar conformation. In order to engender more potent inhibitor against HIV with great activity and less cytotoxicity, a number of _L_-nucleoside have been reported, most of which bear the chemically derivatized pyrimidine and dideoxy _L_-ribose.

Some β-_L_-2′,3′-didehydro-2′,3′-dideoxythymidine (β-_L_-d4T) analogues (Figure 3, **7a**) have been synthesized, all bearing a tether on the C-5 position of the uracil ring, and they were evaluated in vitro for anti-HIV-1 activity [40]. The results revealed that the _L_-d4T derivative, containing 12 methylene units at the 5-position, displayed some activity in the CEM-SS cells (IC_50_ 2.3 µM), probably due to the more lipophilic nature of the nucleoside. Meanwhile, another innovative _L_-d4T derivative, _L_-MCd4T, containing the conformationally rigid methanocarba (MC) nucleoside, was synthesized, and its bicyclo[3.1.0]hexane moiety was restrained to North conformation and the 2′,3′-double bond further reduced the structural flexibility (Figure 3, **7b**) [41]. _L_-MCd4T was found to be a potent anti-HIV-1 inhibitor (EC_50_ 6.76 µg/mL) without significant cytotoxicity, which is comparable to clinical drug ddI [42].

Additionally, some _L_-enantiomers of 2′,3′-dideoxycytidine analogues were reported to selectively inhibit HIV in various cell cultures. For example, _L_-2′,3′-dideoxycytidine, _L_-2′,3′-dideoxy-5-fluorocytidine (_L_-ddC and _L_-FddC, Figure 3, **7c**), _L_-2′,3′-didehydro-2′,3′-dideoxycytidine, and _L_-2′,3′-didehydro-2′,3′-dideoxy-5-fluorocytidine (_L_-d4C and _L_-Fd4C, Figure 3, **7d**) were found to have impressive inhibitory activity but significantly less toxicity when treated in different cells, including rat glioma, lung carcinoma, lymphoblastoid, and skin melanoma cells [43,44,45]. The _L_-enantiomers of 2′,3′-dideoxy-3′-thiacytidine and its 5-fluoro-derivative (_L_-3TC and _L_-FTC, Figure 3, **7e**), when having a sulfur atom in place of the 3′-carbon, were discovered to be a potent inhibitor against HIV-1 in peripheral blood mononuclear cells and were also effective in thymidine kinase-deficient CEM cells. Meanwhile, nontoxicity was observed in human lymphocytes and other cell lines at up to 100 µM [46,47,48].

### 2.4. _L_-Azanucleoside as Anti-HBV Agents

Various azanucleoside analogues have been pioneered, in which the natural carbone atoms in nucleobases are substituted with bioisosteric nitrogens, and these innovative compounds exhibited promising antitumore activity [49,50]. Inspired by this observation, Sartorelli et al. synthesized various dioxolane azanucleosides with _L_-configuration (Figure 4, **8a**–**d**) and bioevaluated them against HBV (Hepatitis B virus) [51]. The synthetic strategies involved the condensation of dioxolane derivative with silyl-protected azacytosine and azathymine. Interestingly, the in vitro HBV activity assay revealed that only (-)-(2S,4S)-1-[2-(hydroxy-methyl)-1,3-dioxolan-4-yl]-5-azacytosine (**8a**) possessed superior activity against HBV (EC_50_ = 0.6 µM), whereas its _D_-analogue was found inactive. No significant antiviral activity was observed in _L_-azathymidine analogues.

### 2.5. _L_-4′-Thionucleosides as Anti-Tumor Agents

The corresponding _L_-4′-thionucleosides (Figure 5, **9a**–**d**) were synthesized by starting from 1,2,3,5-tetra-O-acetyl-4-thio-β-L-ribofuranose [52]. All tested compounds showed a moderate growth inhibitory activity against HTB14 human glioma cells. Notably, compounds **9b** and **9c** exhibited a significant growth stimulatory activity towards NB4 and T47D cells at concentrations 0.78–1.56 µM.

### 2.6. _L_-5-Fluoronucleoside to Treat Leukemia

It has been reported that a type of cytidine analogue, Cytarabine or Ara-C, can be used as an effective chemotherapy against acute myeloid leukemia, acute lymphocytic leukemia, chronic myelogenous leukemia, and non-Hodgkin’s lymphoma [53]. Cytarabine has the arabinoside sugar to mimic the native deoxycytidine, and it is rapidly phosphorylated into cytosine arabinoside triphosphate (Ara-CTP) to interfere with the DNA synthesis in the S phase of the cancer cell cycle [54]. However, the drug resistance of Ara-C is quite common, because the effective Ara-CTP can be easily deaminated to an inactive uridine metabolite by cytidine deaminase (CDA) [55]. To address this issue, an _L_-nucleoside analogue of Ara-C, 5-fluorotroxacitabine (5FTRX) has been developed and activity tested in Acute Myeloid Leukemia (AML) cell lines [56].

5FTRX has an _L_-configuration, containing a fluorinated cytosine nucleobase and dioxolane ring as sugars (Figure 6, **10**). Experimental results suggested that 5-FTRX could also be phosphorylated to its 5′-triphosphate nucleotide, leading to significant DNA chain termination during DNA replication and cancer cell death. In addition, compared to Ara-C, 5FTRX was observed to overcome the drug resistance induced by CDA overexpression to some extent. Meanwhile, no signs of significant toxicity were displayed in a mouse experiment.

### 2.7. LdT as Anti-HBV Agent

The LdT drug, also known as Telbivudine and invented by Novartis Inc., is an FDA-approved (in 2006) anti-viral drug used in the treatment of hepatitis B infection (Figure 7, **11**) [57]. Telbivudine has the _L_-converted structure of native thymidine, and it impairs the HBV DNA replication by leading to chain termination. Clinical trials have fully demonstrated its effect of viral suppression in patients and less viral resistance.

### 2.8. 3TC and FTC to Treat HIV

Lamivudine and Emtricitabine (commonly called 3TC and FTC, Figure 8, **12a****,b**) are _L_-type cytidine analogues both containing oxathiolane rings as sugars. Emtricitabine has an additional Fluoro-modification at the 5-position of its cytosine nucleobase. Both compounds have been FDA approved for the treatment of human HIV and HBV infections, which can be administered individually or in combination with other inhibitors [58]. As the cytidine analogues, 3TC and FTC share the similar mechanism of action, by inhibiting the HIV reverse transcriptase and hepatitis B virus polymerase functions. The lack of 3′-OH groups prevents viral DNA elongation and terminates viral DNA growth. As the non-natural _L_-nucleoside, both drugs were identified as less toxic agents in mitochondrial DNA [59].

### 2.9. _L_-3′-Azido-2′,3′-dideoxypurine Nucleosides as Anti-HIV and Anti-HBV Agents

Inspired by the success of Lamivudine and Emtricitabine and the functional AZT-containing 3′-azido group, the Schinazi lab has prepared various _L_-3′-azido-2′,3′-dideoxypurine nucleosides (Figure 9, **13a****,b**), and evaluated their activity against HIV and HBV [60]. Eleven different _L_-nucleosides were obtained through microwave-assisted optimized transglycosylation reactions. These _L_-nucleoside analogues could be metabolized to corresponding nucleoside 5′-triphosphate compounds in primary human lymphocytes. Weak antiviral activities against HIV-1 and HBV were exhibited, even though no significant toxicity was observed.

### 2.10. _L_-5′-Ethylenic and Acetylenic Modified Nucleosides

The unique 5′-cap structure in mRNA is essential for effective binding to ribosome [61]. Interference with the formation of cap structure could inhibit the replication process and provide the potential strategy for viral treatment. The 5′-cap formation relies on the catalysis by methyltransferases, which has become a popular target for anti-viral drug design [62]. It has been found that many adenosine analogues displayed the interesting antiviral activity by inhibiting S-Adenosyl-_L_-homocysteine (SAH) hydrolase, because SAH hydrolase is a key regulator of many S-adenosyl-_L_-methionine (SAM) dependent biological methylation processes [63]. Various 5′-ethylenic and acetylenic substituted _L_-adenosine derivatives were synthesized (Figure 10, **14a**–**e**), and some of them showed modest inhibition of SAH hydrolase at 100 μM, when tested in the growth of HeLa cells or Bel-7420 cells [64].

### 2.11. _L_-3′-Cyano Modified Nucleosides

Following the similar principle of _L_-d4T and _L_-ddC to treat HIV by inhibiting the viral reverse transcription with reduced toxicity, the Chu lab has developed a series of _L_-nucleoside analogues containing a cyano group at the 3′-position (Figure 11, **a**–**l**). Some of the compounds also contained 2′,3′-unsaturated ribose [65]. The synthesized nucleosides were tested for anti-HIV activity in human PBM cells in vitro, and five of them (derivatives **a**, **b**, **g**, **h** and **j**) showed modest antiviral activity.

### 2.12. _L_-Enantiomer of Ribavirin

Ribavirin is a nucleoside analogue used to treat respiratory syncytia viral infection [66] and HCV infection [67]. It is reported to have the effects of inducing type 1 cytokine bias and enhancing the T cell-demiated immunity in vivo [68]. Structurally, Ribavirin is similar to nucleoside, which has the nucleobase replaced by 1,2,4-triazole-3-carboxamide. Ramasamy et al. have synthesized a series of _L_-nucleoside analogues of Ribavirin and evaluated their activity of stimulating type 1 cytokine and enhancing T cell-demiated immunity [69]. One of the compounds prepared, which had 1,2,4-triazole-3-carboxamide (Figure 12, **17a**) as its nucleobase, was found to be the most uniformly potent compound with interesting immunomodulatry potential.

### 2.13. _L_-Dideoxy Bicyclic Pyrimidine

Zika virus is a mosquito-born flavivirus that can cause the symptoms of fever, rash, joint pain, and red eyes [70]. Zika virus shares the same replication cycle as other flaviviruses, suggesting the potential of using nucleoside analogues as antiviral agents to terminate Zika virus DNA elongation. The Brancale lab has screened a targeted small molecule pool against Zika virus in vitro [71]. Several modified adenosine compounds have been identified to significantly inhibit the virus-induced cytopathic effect. Interestingly, one additional prodrug, **18a**, which exhibited a _L_-nucleoside conformation, was also screened out (Figure 13). This analogue contained _L_-dideoxyribose sugar and a bicyclic pyrimidine as its nucleobase, and it had a potent synergistic effect of inhibiting the vaccinia and measles viruses when applied together with other adenosine phosphoramidate compounds.

Above, we listed some of the _L_-nucleoside analogues with a solid demonstration of their therapeutic activities, which are summarized in Table 1. Besides, there are many other nucleoside-like small molecules designed and evaluated as antiviral therapeutics, which had an _L_-configuration similar to its modified nucleoside, including cyclobutene _L_-nucleoside analogues [72], _L_-erythro-hexopyranosyl nucleosides [73,74], _L_-4′-C-ethynyl-2′-deoxypurine nucleosides [75], _L_-ribo-configured Locked Nucleic Acid [76,77,78,79], pyrrolo, pyrazolo, or imidazo-modified _L_-nucleoside [80], et al. There have also been many methods developed to efficiently synthesize carbocyclic _L_-nucleoside analogues [52,81,82,83], and they have been summarized elsewhere [84,85].

In the future development of _L_-nucleoside as therapeutic molecules, one direction is to design a broad-spectrum antiviral drug essential for rapid and efficient disease treatment. The recent viral outbreaks in the past decade have urged this need. Generally, the development of _L_-nucleosides with broad-spectrum antiviral activities is more challenging because of the different behaviors among viruses, especially after their infections to the host. To design a broad-spectrum antiviral nucleoside, a comprehensive investigation is needed to discover the biological features of multiple viruses and design chemically modified _L_-nucleosides to target these features. In addition, the combinatory treatment using different nucleoside drugs, or a nucleoside with another biological agent, will be necessary to decrease the drug resistance.

## 3. _L_-Nuclei Acid Aptamers to Target Disease-Related Elements

### 3.1. _L_-Aptamers Bind to Small Molecules

In 1996, a 58-mer _L_-RNA aptamer had been isolated from a combinatorial pool to selectively bind to naturally occurring _D_-adenosine [86]. Although the binding affinity was not optimal (K_d_ 1.7 μM), the _L_-RNA aptamer displayed an extraordinary stability in human serum, and it shed light on the potential application of elaborated _L_-RNA ligands as stable monoclonal antibody analogues.

Malachite green (MG) is a popular organic material, frequently used as a dye and an antimicrobial in aquaculture and the food industry [87]. The wide range of toxicological effects of MG have been identified, including carcinogenesis, mutagenesis, chromosomal fractures, teratogenicity, and respiratory toxicity [88]. Therefore, the sensitive and rapid detection as well as the clearance of MG are critical. The Huang lab has developed an _L_-RNA aptamer to selectively bind MG [89]. Since malachite green as the selection target is a planar molecule without characteristic chirality, both _L_-RNA and _D_-RNA aptamers can bind to it with the identical affinities. However, the _L_-aptamer exhibited excellent stability and remained durably intact in the standard buffer. This result suggested the promising potential of this _L_-aptamer as a sensitive and reliable analysis tool in environment monitoring and animal tissue examination.

Clearly, when detecting small molecules without chirality, _L_-RNA and _D_-RNA aptamers sharing the same sequence should have the same binding affinities. Malachite green is just one example. Following this principle, several labs have directly replaced the reported _D_-aptamers with _L_-aptamers to target nonchiral targets, including Hemin, cationic porphyrin [90], and ethanolamine [91]. As expected, _L_-aptamers displayed greatly improved biostability and identical binding affinity as _D_-aptamers. Interestingly, the chiral molecule of natural ATP was also bound to an _L_-aptamer in the same manner as a _D_-aptamer, since the intermolecular interactions are independent of chiral atoms in ATP [90].

### 3.2. _L_-Aptamers Bind to RNA Motifs

When targeting small RNA motifs, _L_-aptamers were commonly selected as binding ligands to selectively target disease-relevant noncoding RNAs. SELEX is the main approach of aptamer identification. In the practical isolation, a wild-type _D_-aptamer is first identified against the synthetic mirror-image of the intended target from the random DNA/RNA libraries. Secondly, the isolated aptamers are converted to _L_-oligonucleotides binding to the wild-type targets. For instance, different L-RNA aptamers have been obtained by in vitro selection against the precursor microRNAs, which are oncogenic precursor of mature microRNAs. The successful examples include pre-miR-155 aptamers [92,93] and pre-miR-19a aptamers [94]. These _L_-DNA or _L_-RNA aptamers were observed to have exceptional affinities and intrinsical stability. They can bind to pre-microRNA targets through unique tertiary structural features and block the classic Dicer-mediated processing of pre-microRNAs to generate mature microRNAs. Given their particularities, these _L_-aptamers might be further explored and applied as future miRNA inhibitors and cancer therapies.

In addition, _L_-RNA aptamer has been developed successfully as potential anti-HIV agent. A 43mer _L_-RNA aptamer was selected to bind the natural HIV-1 trans-activation responsive (TAR) RNA with a K_d_ of 100 nM [95]. The cross-chiral intermolecular bridging highly depends on the tertiary interactions and the characteristic six-nucleotide distal loop of the TAR element. The _L_-aptamer has the therapeutic promise to interfere with the TAR-tat protein complex formation in HIV, thereby interrupting the normal TAR function as well as the viral RNA genome expression.

The Kwok lab reported two different _L_-RNA aptamers, which can target native RNA G-quadruplexes (G4s) with the K_d_ of 75 and 99 nM, respectively [96,97]. G4s is one featured RNA structural motifs, containing multiple G-quartet planes that can be stabilized by monovalent ions [98]. The structure is important for its interaction with G4-binding proteins, which modulate telomerase activity and regulate gene expression [99]. Therefore, the development of these _L_-RNA aptamers is of great significance in cancer diagnosis and treatment.

Recently, a novel _L_-DNA aptamer was successfully identified as targeting the stem-loop II-like motif (s2m), which is a highly conserved structure at the 3′-UTR region of the RNA genome of severe acute respiratory syndrome coronavirus 2 (SARS-CoV-2) [100]. The 41-nucleotide s2m element is discovered in coronavirus, astrovirus, and equine rhinovirus genomes, and it plays a critical role in viral replication and packaging [101]. Therefore, s2m structure is an effective and popular target in anti-viral drug design. With the optimal binding affinity and selectivity, as well as the outstanding stability against nucleases, this _L_-DNA aptamer provides the opportunity to offer a new therapeutic molecule to end this unprecedented pandemic.

### 3.3. _L_-Aptamers Bind to Amino Acids and Proteins

In 1996, Nolte et al. selected a stable 38-mer _L_-RNA oligoribonucleotide to bind to _L_-arginine and a short peptide containing the basic region of HIV-1 Tat-protein [102]. _L_-RNA was observed to own the exceptional stability against human serum. By binding to HIV Tat-protein and blocking the regulatory function of Tat, this _L_-aptamer had the potential to interfere with the transcription of the HIV genome as an innovative anti-HIV drug.

The Peyrin lab has applied the mirror-image strategy to design the _L_-RNA aptamer used in chiral stationary phase (CSP) against different natural amino acid residues [103,104,105,106]. The successful targets included _L_-arginine, _L_-tyrosine, and _L_-histidine. The selected 40–60 nucleotides aptamers displayed prominent resistance to the enzymatic degradation and resolved the different racemates well. This _L_-RNA aptamer-based CSP is an effective approach for the chromatographic separation of the enantiomers of amino acid residues.

As the mirror-image-structured oligonucleotides, considerable efforts have been delivered to develop _L_-aptamers to bind and antagonize the disease-relevant target. The most famous instance is the registered trademark of Spiegelmer, belonging to NOXXON Pharma AG. Selection of _L_-aptamers against proteins follows the similar procedure as those against microRNAs. With the non-natural chirality, Spiegelmers has a abiding survival time in biological fluids [107]. Different Spiegelmer aptamers have been isolated to target various pharmacological molecules, and several of them are in different preclinical animal models and clinical phases.

Complement factor C5a is a potent proinflammatory mediator that contributes to the pathogenesis of a variety of diseases, including autoimmune diseases, acute inflammatory responses, and systemic inflammatory response syndrome caused by ischemia/reperfusion injuries, trauma, or systemic infection in sepsis [108,109]. It has been discovered that neutralizing C5a with binding ligands significantly prevents multiorgan failure and improves survival as well as prevents the exhaustion and dysfunction of neutrophils [110]. Accordingly, effective therapeutic candidates of Spiegelmers, NOX-D20 [111], and NOX-D21 [112] have been screened out. These PEGylated biostable mirror-image mixed (_L_-)RNA/DNA aptamers bind to mouse and human C5a with pM affinities and demonstrated promising effects on attenuating inflammation and organ damage, preventing the breakdown of the vascular endothelial barrier, and improving survival. Moreover, the crystal structure of this therapeutic Spiegelmer complexed with mouse C5a has been determined, illustrating the molecular insights into the _L_-aptamer binding basis [113].

Another example of Spiegelmer, NOX-A12, has been identified to inhibit Stromal Cell Derived Factor 1 (SDF-1), which binds to CXCR4 chemokine receptor and mediates hematopoietic cell migration to bone marrow stroma. NOX-A12 could inhibit SDF-1 in vitro and in vivo, against BCR-ABL- and FLT3-ITD-dependent leukemia cells. NOX-A12 itself, or in synergistic application with the Tyrosine Kinase Inhibitor drug, significantly suppressed the SDF-1 induced cell migration and reduced the leukemia burden in mice to a greater extent [114]. Besides, NOX-A12, together with another Spiegelmer mNOX-E36, was evaluated in a syngeneic mouse model of intraportal islet transplantation and in a diabetes induction model containing multiple low doses of streptozotocin (MLD-STZ) [115,116]. Both _L_-aptamers significantly improved islet engraftment, decreased the recruitment of inflammatory monocytes, and attenuated diabetes progression. The crystal structure of aptamer mNOX-E36 bound to pro-inflammatory chemokine CLL2 was also determined to be 2.05 Å, and it unveiled the structural rationale of _L_-aptamer targeting certain chemokines [117].

The sphingolipid S1P (sphingosine 1-phosphate) is involved in a number of pathophysiological conditions [118]. It is well understood that S1P plays a vital role by defining the components of the plasma membrane, as well as in various significant cell signaling pathways and physiological processes such as cell migration, survival, and proliferation, cellular architecture, cell-cell contacts and adhesions, inflammation and immunity, and tumorogenesis and metastasis [119]. It is obvious that regulating the S1P synthesis and degradation will be critical to govern a variety of cellular and physiological processes. Another Spiegelmer, NOX-S93, has been employed in vitro to selectively bind to S1P with a K_d_ of 4.3 nM [120]. The results provided evidence that NOX-S93 inhibited S1P-mediated β-arrestin recruitment and intracellular calcium release in a Chinese hamster ovary cell model. Furthermore, the pro-angiogenic activity of S1P was also dramatically blocked by NOX-S93, in a cellular angiogenesis assay employing primary human endothelial cells. It suggested the therapeutic potential of NOX-S93 when neutralizing pharmacological S1P as a promising approach related to angiogenesis.

Calcitonin gene-related peptide (CGRP) plays a major role in the pathogenesis of migraine and other primary headaches [121]. CGRP receptor inhibitors and anti-CGRP antibodies have been demonstrated to be therapeutically effective in migraines. Fischer et al. have selected an _L_-aptamer NOX-C89 to target CGRP, and single neuron cell activity in mouse model has been assayed when infusing ascending doses of NOX-C89 [122]. The abolishment of heat-induced neuronal activity could be observed after infusing certain amount of NOX-C89, which indicated the potential of this _L_-aptamer to control spinal trigeminal activity.

High-mobility group A1 (HMGA1) proteins are usually overexpressed in human malignancies and promote anchorage-independent growth and epithelial-mesenchymal transition [123]. Therefore, HMGA1 is considered a potential therapeutic target in pancreatic cancer treatment. Spiegelmers NOX-A50 and NOX-f33 have been successfully selected in vitro as HMGA1 binders [124]. The in vivo assay using a xenograft mouse model with a pancreatic cancer cell line showed the effectiveness of NOX-A50 to significantly reduce tumor volumes.

In order to select the ligand to inhibit and study the RNase function in biological systems, the Joyce lab has identified an _L_-RNA aptamer targeting the intact barnase enzyme [125]. The aptamer could specifically bind to the barnase with an affinity of ∼100 nM and function as a competitive inhibitor of the enzymatic cleavage of _D_-RNA substrates. It provided notable implication for application in molecular diagnostics and therapeutics.

## 4. _L_-Nucleic Acid Nanoparticles as Drug Delivery Tools

As the enantiomer of natural DNAs/RNAs, _L_-type nucleic acids composed of nucleobases, phosphate linkages, and _L_-(deoxy)ribose sugars (Figure 14A,B) retain the identical structural and physical properties, including aqueous solubility, Watson–Crick base pairing, secondary structural conformation, and duplex thermal stability. Due to the chiral inversion, _L_-DNAs/RNAs are orthogonal to the stereospecific environment of biology and are unable to interact with native nucleases, leading to complete resistance to nuclease degradation and, substantially, avoiding immune response and nonspecific therapeutic toxicity [126].

Since the geometry and periodicity of _L_-DNA/RNA structures were the same as that of their _D_-counterparts, the well-established design principles and protocols on nucleic acid nanostructures can be directly employed to _L_-DNAs/RNAs without further modification. Besides, as above mentioned, _L_-DNA/RNA molecules have the essential features of selectively binding to molecular targets, after elaborative design and manipulation [127,128]. These observations bring out appealing opportunities for in vivo medical applications of _L_-DNA/RNA nanostructures.

Early in 2009, Lin et al. constructed a series of short _L_-DNA architectures that could readily self-assemble into various well-defined 1D and 2D nanostructures with opposite chirality, including a 1D finite four arm junction (J1) (Figure 14C), 2D infinite extended _L_-DNA nanotubes (Figure 14D), and large infinite two-dimensional nanoarrays (Figure 14E) [129]. These supramolecular _L_-DNA self-assemblies enriched the library of unnatural nucleic acids, presenting identical physical properties and exceptional nuclease resistance to survive in a biological environment.

A wireframe _L_-DNA tetrahedron (_L_-Td) has also been demonstrated by Kim et al. in 2016 [130]. This work was based on the native _D_-DNA tetrahedra (_D_-Td), designed with a tetrahedral shape formed by the precise organization of complementary base-pairs [131]. These _D_-DNA tetrahedra possessed cell-penetrating properties, which enabled the nanoconstructs to effectively carry various bioactive molecules as an intracellular delivery machine in the absence of traditional transfection agents [132,133]. The enantiomeric _L_-DNA-assembled tetrahedra have shown enhanced cellular uptake, biostability and pharmacokinetics in vivo compared to natural DNA tetrahedra. Due to the extraordinary properties of _L_-Td, it has exhibited a highly effective tumor-specific delivery of doxorubicin (DOX). Additional application of _L_-Td includes the streptavidin-Td (STV-Td) hybrid developed as a powerful carrier for intracellular delivery of various enzymes, such as the anti-proliferative proteins/enzymes, leading to significant suppression of tumor growth [134]. Therefore, the STV-Td hybrid potentially enlarged the inventory pool of protein therapeutics for drugging undruggable molecular cancer-relevant targets [135]. However, the restriction on length and size of _L_-DNAs/RNAs due to the inability of syntheses by traditional enzymatic polymerization largely limits the potential applications of L-nanodevices.

Recent work in the Sczepanski lab has reported a novel methodology taking advantage of peptide nucleic acids (PNAs), which have no chirality but can hybridize to DNA and RNA, thus enabling two toehold-mediated strand displacement [136,137]. The strand displacement utilized a DNA:PNA heteroduplex for the sequence-specific interfacing of the two enantiomers of DNAs. This work notably lays the foundation for an interfacing heterochiral circuit employed as diagnostic and/or therapeutic nanodevices and has great potential in constructing biorthogonal nanoparticles comprised of diverse nucleic acids in living cells and organisms.

## 5. Conclusions and Outlook

Nucleic acid therapeutics is a rapidly emerging field of biotechnology for disease treatment, especially after the unprecedented and prompt development of mRNA vaccines during the pandemic [138]. As the nucleic acid substitutes containing unusual chirality, _L_-nucleic acids are featured with unique properties, which can help to address some long-lasting bottlenecks in this area. The rapid growth of _L_-nucleic acids and _L_-nucleoside therapeutics can solve the conventional problems of stability in vivo, delivery efficiency and, possibly, immunogenicity. These innovative molecules have created a versatile and powerful platform, which has unlimited potential to meet future clinical needs and care for many diseases.

Whereas, there is room to further improve _L_-nucleoside and _L_-nucleic acids therapeutics. The discovery and development of _L_-nucleoside molecules with broad-spectrum antiviral activities will be a primary target in the coming decades. Moreover, unlike the biotechnology of using native _D_-antisense, siRNA, microRNA, and aptamer drugs in clinical treatment, there are not many chemically modified _L_-nucleic acids available. Most of the _L_-nucleic acid aptamers and nanoparticles under development are based on unmodified _L_-nucleotides. Though the primary intention of modifying native DNA/RNA drugs was to enhance their stability against serum, this concern is negligible for _L_-nucleic acids; however, versatile modifications on nucleobases and backbones can also offer unique chemical and structural features [139]. For example, the bulky derivatizations on (deoxy)riboses can increase the thermostabilities, and some hydrophobic substitutions would boost the binding capacities of the aptamers. If _L_-nucleic acids are equipped with diverse chemical moieties, it will certainly offer more expanded functions to benefit their application. There have been a few examples that chemically derivatized _L_-DNAs/RNAs have more optimal properties [94,140,141].

Secondly, the preparation of _L_-oligonucleotides is restricted by the efficiency of solid-phase synthesis. Phosphoramidite chemistry is the main approach to synthesize _L_-oligonucleotides, and it is inefficient and costly when longer strands are needed, especially in the case of RNA. After billions of years of evolution, there is a lack of competent polymerases to accurately recognize and readily amplify _L_-nucleic acids. The efforts are now delivered to engineer non-natural polymerases, either via the chemical syntheses of entirely modified _D_-polymerases [142,143,144,145] or through searching for the ancient polymerase, which can interact with both enantiomeric nucleic acids [146]. It is worth noting, some labs are also focused on developing artificial ribosomes, which contain mutated 23S ribosomal RNA [147,148,149]. The purpose is to engender the ribosome to carry on the _D_-peptide elongation when the tRNA substrate is aminoacylated with _D_-amino acid residue. This evolved ribosome would be applied in the translation process to efficiently produce mirror-image _D_-enzymes, such as polymerases, ligases, helicases, and other _L_-nucleic acid-contacting enzymes, to eventually realize the high-throughput syntheses of _L_-nucleic acids. If accomplished, it could generate great potential in the application of mirror-image biological molecules, from the bench of basic science to the bedside of clinical disease treatment.

## Figures and Tables

**Figure 1 genes-13-00046-f001:**
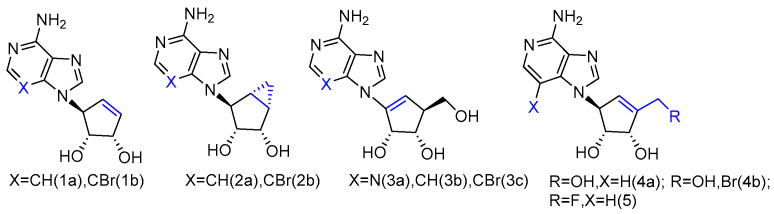
_L_-like analogues of carbocyclic nucleoside neplanocin.

**Figure 2 genes-13-00046-f002:**
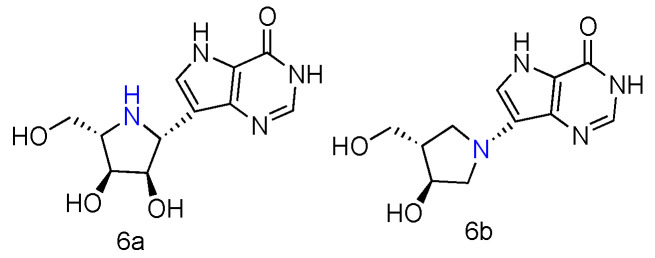
Structures of _L_-ImmH (**6a**) and _L_-DADMe-ImmH (**6b**).

**Figure 3 genes-13-00046-f003:**
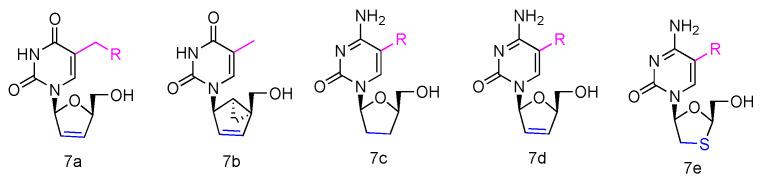
Structures of anti-HIV _L_-pyrimidine nucleoside analogues.

**Figure 4 genes-13-00046-f004:**
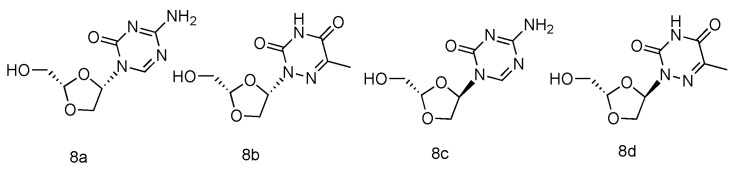
Structures of _L_-azanucleoside as an anti-HBV agent.

**Figure 5 genes-13-00046-f005:**
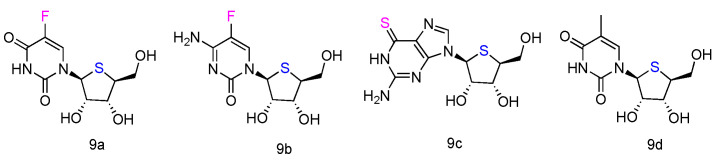
Structures of _L_-4′-thionucleosides.

**Figure 6 genes-13-00046-f006:**
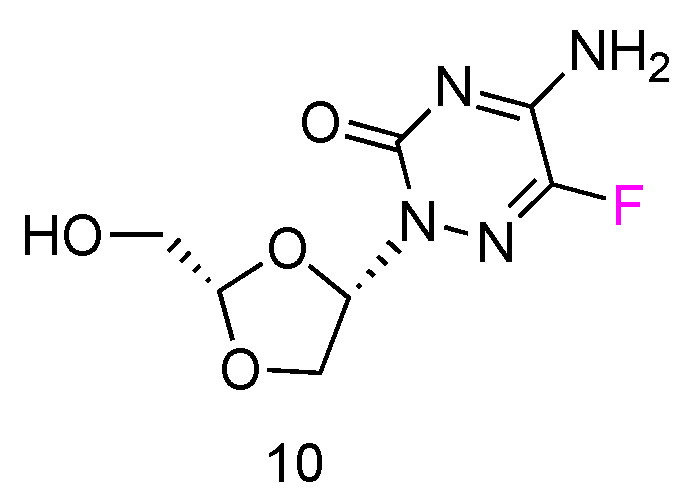
Structures of _L_-5-fluorotroxacitabine.

**Figure 7 genes-13-00046-f007:**
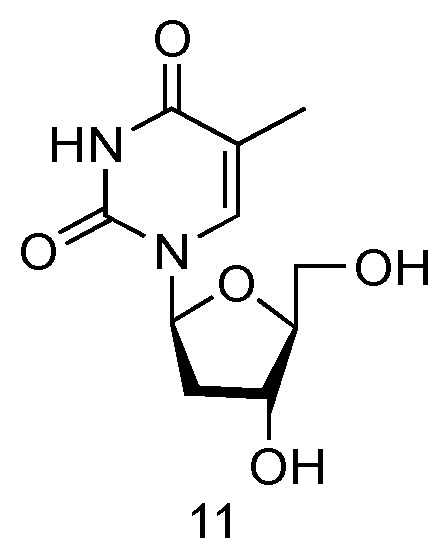
Structures of LdT.

**Figure 8 genes-13-00046-f008:**
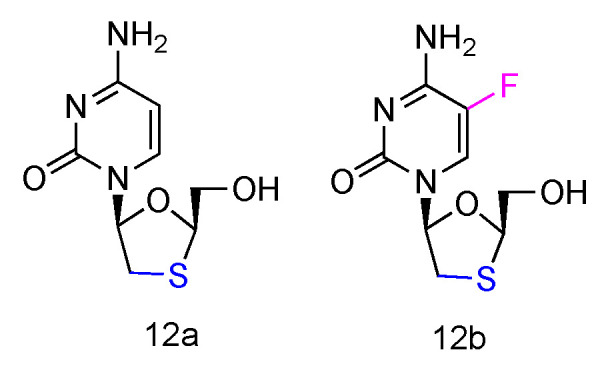
Structures of Lamivudine and Emtricitabine.

**Figure 9 genes-13-00046-f009:**
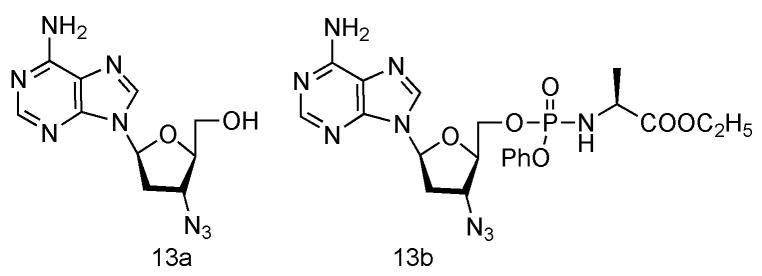
Structures of _L_-3′-azido adenosine and its phosphoramidate derivatives.

**Figure 10 genes-13-00046-f010:**
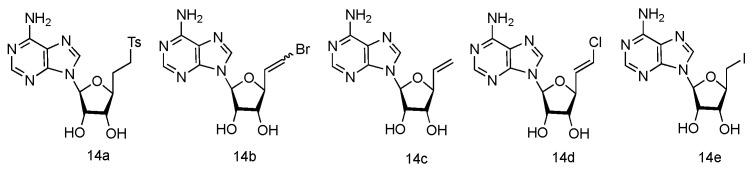
Structures of _L_-5′-ethylenic and acetylenic nucleosides.

**Figure 11 genes-13-00046-f011:**
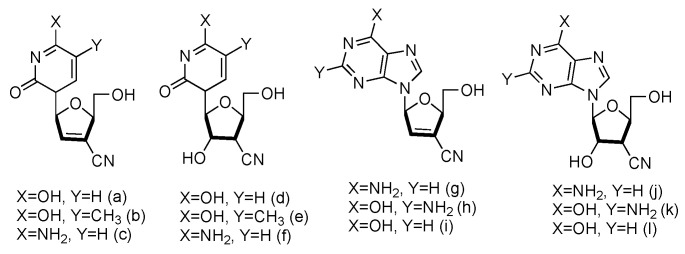
Structures of _L_-3′-cyano modified nucleosides.

**Figure 12 genes-13-00046-f012:**
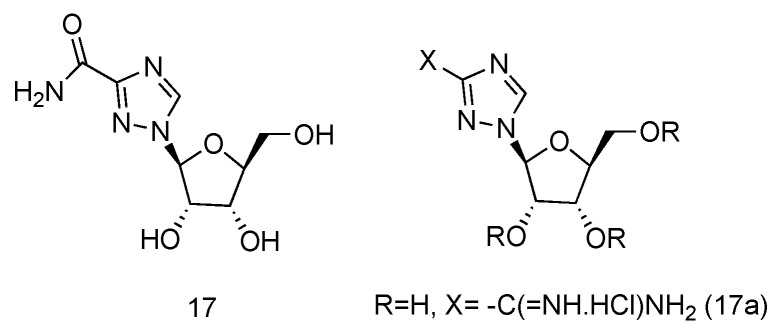
Structure of _L_-Ribavirin (**17**) and its derivative (**17a**).

**Figure 13 genes-13-00046-f013:**
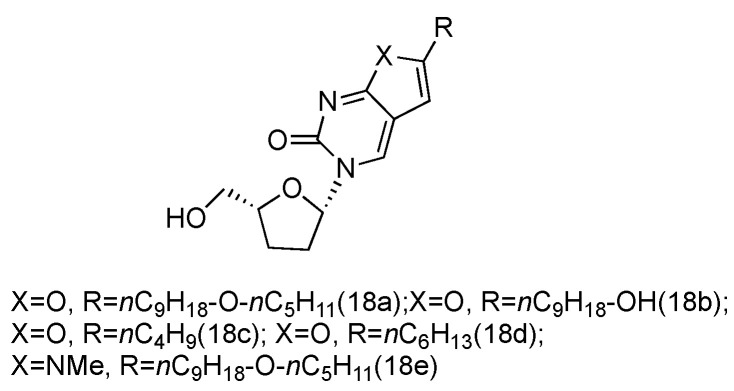
Structures of _L_-dideoxy bicyclic pyrimidine.

**Figure 14 genes-13-00046-f014:**
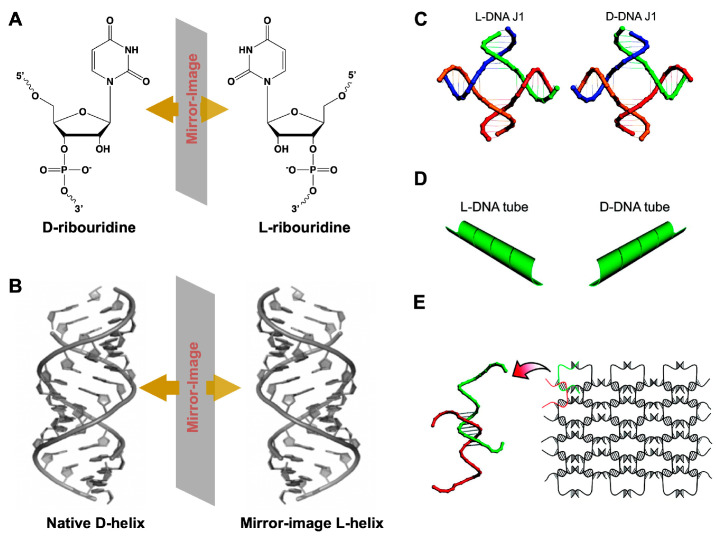
(**A**) Chemical structures of _D_-ribouridine, _L_-ribouridine, and (**B**) duplex of native _D_-DNA and _L_-DNA. (**C**) Structural models of _L_- and _D_-DNA J1 molecules. (**D**) The opposite chirality of the L-(left-column) and _D_-(right column) DNA nanotubes revealed. (**E**) Schematic drawings of two-dimensional nanoarrays self-assembled from L-DNA. (**C**–**E**) reprinted with permission from [126]. Copyright 2009 American Chemical Society.

**Table 1 genes-13-00046-t001:** List of synthetic _L_-nucleoside analogues for antiviral and antitumor evaluations.

Evaluated Analogues	Nucleobase	Sugar	Activity
Neplanocin analogues	Adenine analogue	Cyclopentenol analogue	Anti-norovirus and Ebola virus.
Immucillin analogues	Inosine analogue	Iminoribitol analogue	Treatment of T-cell acute lymphoblastic leukemia
_L_-Pyrimidine analogues	Modified pyrimidine	Dideoxy ribose	Anti-HIV
_L_-Azapyrimidine analogues	Azapyrimidine	Dioxolane	Anti-HBV
_L_-4′-Thionucleosides	5-Fluoropyrimidine, 6-thioguanine	4′-Thioribose	Anti-tumor
5-Fluorotroxacitabine (5FTRX)	5-Fluorocytosine	Dioxolane	Treatment of leukemia
LdT	Thymidine	Deoxyribose	Anti-HBV
Lamivudine and Emtricitabine	Cytidine and fluoro-cytidine	Oxathiolane rings	Anti-HIV and Anti-HBV
_L_-3′-Azido-2′,3′-dideoxypurine nucleosides	Modified purines	3′-Azido-2′,3′-dideoxyribose	Anti-HIV and Anti-HBV
_L_-5′-Ethylenic and acetylenic modified nucleosides	Adenine	5′-Ethylenicand acetylenic ribose	Inhibit SAH for virus treatment
_L_-3′-Cyano nucleosides	Modified purine and pyrimidine	3′-C-cyano-2′,3′-unsaturated and 3′-C-cyano-3′-deoxyribose	Anti-HIV
_L_-Isomer of ribavirin	1,2,4-triazole-3-carboxamide	Ribose	Induce type 1 cytokine for viral treatment
_L_-Dideoxy bicyclic pyrimidine	Bicyclic pyrimidine	Dideoxy ribose	Anti-ZIKA virus

## Data Availability

Not applicable.

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
