# Peer review of "Advances in Therapeutic L-Nucleosides and L-Nucleic Acids with Unusual Handedness"

_genes, 2021, doi:10.3390/genes13010046_

Round 1

Reviewer 1 Report

The manuscript is a good review on the therapeutic applications of nucleosides and nucleotide analogues composed of L-enantiomers. The review focuses on the one hand on L-DNA / RNA compounds (L-nucleosides) due to their resistance to degradation by nucleases and little or no toxicity and immunoginicity and on the other hand on aptamers (L-aptamers) as ligands. with functions equivalent to monclonal antibodies. Finally, the application in the formation of nanoparticles (L-nanoparticles) to administer drugs is contemplated. The applications are mainly antivial and antitumor strategies, molecular targets of diseases and drug delivery.

Although the review is interesting, there are some issues that must be taken into consideration.

1) About the title: Since the article mainly reviews different aspects of L-nucleic acids (including L-nucleoside, L-aptamer and L-nanoparticle), the term "L-nucleic acids" should appear in the title

2) About the section of Nucleoside analogs as therapeutic antiviral agents: a) the title of this section does not correspond well to the content, since in addition to antiviral therapies, it also includes antitumor therapies. Therefore, either the title is changed including both therapies, or a separate section is made for each therapy; b)  In the figures in this section, it would be of interest to include as a reference the natural structure of the D-enentiomer, in order to facilitate seeing the structural differences with the proposed L-analogs;  c) line 232: there is a typographical error: "2.8.3 TC and FTC ..." should change to "2.8 3TC and FTC"

3) About the section, L-Nucleic acid nanoparticles as drug delivery tools: line 483: in figure 14 (B), is correct the legend ? or is DNA duplex?

Author Response

“The manuscript is a good review on the therapeutic applications of nucleosides and nucleotide analogues composed of L-enantiomers. The review focuses on the one hand on L-DNA / RNA compounds (L-nucleosides) due to their resistance to degradation by nucleases and little or no toxicity and immunoginicity and on the other hand on aptamers (L-aptamers) as ligands. with functions equivalent to monclonal antibodies. Finally, the application in the formation of nanoparticles (L-nanoparticles) to administer drugs is contemplated. The applications are mainly antivial and antitumor strategies, molecular targets of diseases and drug delivery.

Although the review is interesting, there are some issues that must be taken into consideration.

About the title: Since the article mainly reviews different aspects of L-nucleic acids (including L-nucleoside, L-aptamer and L-nanoparticle), the term "L-nucleic acids" should appear in the title.”

  • We thank the reviewer for reviewing our manuscript, the appreciation of our manuscript and all the comments. We have added L- in the title.

  • “About the section of Nucleoside analogs as therapeutic antiviral agents: a) the title of this section does not correspond well to the content, since in addition to antiviral therapies, it also includes antitumor therapies. Therefore, either the title is changed including both therapies, or a separate section is made for each therapy;

  • We added “antitumor” to the title of this section.

  • “b) In the figures in this section, it would be of interest to include as a reference the natural structure of the D-enentiomer, in order to facilitate seeing the structural differences with the proposed L-analogs;” 

  • We tried to add D-nucleosides’ structures there. But since there are multiple structures in each figure in this section, and these structures are not completely the same, it seemed to us very distracting and easy to confuse the readers, after adding these D-enantiomers. Therefore, we decided to keep the current figures.

  • “c) line 232: there is a typographical error: "2.8.3 TC and FTC ..." should change to "2.8 3TC and FTC"

  • Thank you for bringing this out. We have corrected it.

5) “About the section, L-Nucleic acid nanoparticles as drug delivery tools: line 483: in figure 14 (B), is correct the legend ? or is DNA duplex?”

  • This duplex in Figure 14 is a DNA, and its mirror-image molecule. We have corrected the legend.

Reviewer 2 Report

This paper deals with review of publications about three different aspects of L-nucleic acid therapies, including pharmacological L-nucleosides. 

The authors showed many of synthetic L-nucleoside analogues and they briefly describe the aplications of this nucleoside analogs as antiviral agents. 

While the title of the paper and abstract are interesting, I have a problem with rewiev of this paper, but authors have not clear written what this review was about to advance the sciences of nucleosides and their appilacion in vivo. What is the new in this paper? What are conclusions of this paper?

Please change a small mistakes in the paper, for example in line 194.

Author Response

“This paper deals with review of publications about three different aspects of L-nucleic acid therapies, including pharmacological L-nucleosides.

The authors showed many of synthetic L-nucleoside analogues and they briefly describe the applications of this nucleoside analogs as antiviral agents.

While the title of the paper and abstract are interesting, I have a problem with review of this paper, but authors have not clear written what this review was about to advance the sciences of nucleosides and their application in vivo. What is the new in this paper? What are conclusions of this paper?

Please change a small mistake in the paper, for example in line 194.”

  • We thank the reviewer for reviewing our manuscript and all the comments. This is a review article that we try to summarize the current progress in mirror-image nucleic acid-based therapeutic development. For the perspective of L-nucleoside, we have added one paragraph in the end of Section 2. Like the D-nucleoside drugs, the development of L-nucleosides as antiviral and antitumor agents needs medicinal chemistry and biochemical research. We couldn’t discuss too many details here because of the page limit.

  • We have corrected the mistake in line 194.